# Genetic Regulation of Tryptase Production and Clinical Impact: Hereditary Alpha Tryptasemia, Mastocytosis and Beyond

**DOI:** 10.3390/ijms22052458

**Published:** 2021-02-28

**Authors:** Bettina Sprinzl, Georg Greiner, Goekhan Uyanik, Michel Arock, Torsten Haferlach, Wolfgang R. Sperr, Peter Valent, Gregor Hoermann

**Affiliations:** 1Ludwig Boltzmann Institute for Hematology and Oncology at the Hanusch Hospital, Center for Medical Genetics, Hanusch Hospital, 1140 Vienna, Austria; bettina.sprinzl@oegk.at (B.S.); goekhan.uyanik@oegk.at (G.U.); 2Center for Medical Genetics, Hanusch Hospital, 1140 Vienna, Austria; 3Department of Laboratory Medicine, Medical University of Vienna, 1090 Vienna, Austria; georg.greiner@meduniwien.ac.at; 4Ludwig Boltzmann Institute for Hematology and Oncology, Medical University of Vienna, 1090 Vienna, Austria; wolfgang.r.sperr@meduniwien.ac.at (W.R.S.); peter.valent@meduniwien.ac.a (P.V.); 5Ihr Labor, Medical Diagnostic Laboratories, 1220 Vienna, Austria; 6Medical School, Sigmund Freud Private University, 1020 Vienna, Austria; 7Department of Hematology, APHP, Pitié-Salpêtrière-Charles Foix University Hospital and Sorbonne University, 75013 Paris, France; michel.arock@aphp.fr; 8Centre de Recherche des Cordeliers, INSERM, Sorbonne University, Cell Death and Drug Resistance in Hematological Disorders Team, 75006 Paris, France; 9MLL Munich Leukemia Laboratory, 81377 Munich, Germany; torsten.haferlach@mll.com; 10Department of Internal Medicine I, Division of Hematology and Hemostaseology, Medical University of Vienna, 1090 Vienna, Austria

**Keywords:** tryptase, hereditary alpha tryptasemia, mastocytosis

## Abstract

Tryptase is a serine protease that is predominantly produced by tissue mast cells (MCs) and stored in secretory granules together with other pre-formed mediators. MC activation, degranulation and mediator release contribute to various immunological processes, but also to several specific diseases, such as IgE-dependent allergies and clonal MC disorders. Biologically active tryptase tetramers primarily derive from the two genes *TPSB2* (encoding β-tryptase) and *TPSAB1* (encoding either α- or β-tryptase). Based on the most common gene copy numbers, three genotypes, 0α:4β, 1α:3β and 2α:2β, were defined as “canonical”. About 4–6% of the general population carry germline *TPSAB1*-α copy number gains (2α:3β, 3α:2β or more α-extra-copies), resulting in elevated basal serum tryptase levels. This condition has recently been termed hereditary alpha tryptasemia (HαT). Although many carriers of HαT appear to be asymptomatic, a number of more or less specific symptoms have been associated with HαT. Recent studies have revealed a significantly higher HαT prevalence in patients with systemic mastocytosis (SM) and an association with concomitant severe Hymenoptera venom-induced anaphylaxis. Moreover, HαT seems to be more common in idiopathic anaphylaxis and MC activation syndromes (MCAS). Therefore, *TPSAB1* genotyping should be included in the diagnostic algorithm in patients with symptomatic SM, severe anaphylaxis or MCAS.

## 1. Mast Cells and Mast Cell Diseases

In 1877, Paul Ehrlich first mentioned the term “mast cells” (MCs) based on their unique cytoplasmic metachromatic granules after staining with basic dyes, and the assumption that these granules are a product of cell overfeeding [1,2]. Today, we know that these characteristic granules are full of enzymes and other mediators, which are all produced by MCs themselves, and that the granular mediators of MCs play an indispensable role in cellular processes of our immune system [3,4,5]. MCs originate from bone marrow-derived CD34+/CD117+/CD13+ human pluripotent stem and progenitor cells, with the stem cell factor (SCF) being the principal growth factor [3,6,7,8,9,10]. The SCF-dependent growth and differentiation of human MCs are regulated by several additional cytokines, such as IL-4 and IL-6, and other factors [11,12,13,14]. After reaching the mature state, resident MCs can be found within vascularized tissues, where they are involved in host responses in innate and acquired immunity [3,14,15]. Activation of MCs is mainly initiated through the high-affinity immunoglobulin E receptor (FcϵRI) and the receptor tyrosine kinase protein KIT (also known as CD117 or stem cell factor receptor, SCFR). Both receptors are expressed abundantly in human MCs [5,7,8,9,14,15,16,17,18,19]. The receptor-dependent MC activation leads to an initiation of downstream signal transduction cascades, and subsequently to cell degranulation and the secretion of preformed and newly formed mediators (e.g., tryptase, histamine, heparin, serotonin, chymases, cathepsins, growth factors, cytokines, chemokines, leukotrienes and lipid mediators) in MCs [3,8,14,15,20,21,22]. MCs are believed to survive and regranulate after degranulation [21] and can reside in tissues as long-lived immune cells for many months or even years [3,15,23].

MC activation can be induced by several factors and agents, for example, as triggers of certain immune responses, such as IgE-dependent activation by allergens or complement-mediated activation, but also by physical factors (heat and trauma), endogenous mediators (cytokines), pathogens, venoms and toxins [3,14]. The secretion of tryptase and other MC-dependent compounds is observed in various physiological processes, such as inflammation, host defense, adaptive immune response, remodeling of tissue architecture (tissue repair, wound healing), vasodilation, modulation of permeability, angiogenesis and fibrosis [3,24,25,26,27,28,29,30,31,32]. MC degranulation and mediator release have specifically been described in certain clinical settings and diseases, such as asthma, allergy, anaphylaxis, autoimmune diseases (e.g., multiple sclerosis, rheumatoid arthritis and type-1 diabetes) and MC-related disorders (mastocytosis and MC activation syndromes) [3,33,34,35,36,37,38,39].

## 2. Biological Functions of Human Tryptase

Experiments in the early 1950s and 1960s first described a trypsin-like enzyme activity in MCs [40,41,42,43]. Further research revealed that tryptase is the predominant enzyme present in these cells. Tryptase is a tetrameric neutral serine protease primarily expressed by tissue MCs and far less abundantly in blood basophils. The human tryptase amount per MC is ~11 (in lung MCs) and ~35 pg (in skin MCs), and can reach up to 25% of the total cellular protein content in MCs [44,45]. In comparison, the tryptase amount in peripheral blood basophils is only ~0.05 pg/cell [46,47]. Tryptase is packaged and stored in the secretory granules of MCs together with other mediators, and is released upon degranulation after cell activation [45,48,49]. In addition, some of the tryptases, especially the pro-alpha and pre-beta forms of the enzyme, are constantly released from MCs, independent of cell activation (basal secretion). The various forms of tryptase expressed in MCs are described in detail below.

Tryptase serine protease activity on peptide substrates is involved in a number of physiologic processes in various organ systems. Major known substrates that are inactivated upon beta-tryptase exposure include, among others, fibrinogen [50,51], fibronectin [52,53], type VI collagen [54], and high-density lipoprotein (HDL) [55]. Tryptase is also involved in the activation, proliferation and migration of various mesenchymal cells, including endothelial cells, and fibroblasts in various physiologic and pathologic conditions [56,57], in part via the upregulation of various effector molecules, i.e., adhesion proteins (ICAM-1), growth factors (TNF-α) and cytokines (IL-1, IL-6, IL-8 and stem cell factor), and through interaction with protease activated receptors (PARs—especially PAR2) [57,58,59,60,61,62]. Furthermore, the effects of tryptases on coagulation, connective tissue metabolism and smooth muscle cell contractility have been described [63,64,65]. The biological function of tryptase is associated with the proteolysis of peptide substrates via polypeptide cleavage of the carboxy-terminal domain of arginine and lysine residues, or indirect proteolysis mediated by downstream molecules (e.g., PARs). Enzymatically active tryptase tetramers have a molecular weight of approximately ~135 kDa, and consist of four monomer subunits (~30–33 kDa each) [44,46,47,66,67].

In resting tissue MC, tryptase is present in two different forms—monomeric pro-α- and pro-β-tryptase and mature tetrameric α- and β-tryptase. Monomeric pro-α- and pro-β-tryptase are spontaneously and constitutively secreted into tissue and diffuse into the systemic circulation. They are the predominant form of tryptase in serum in the absence of systemic MC activation [68,69]. Mature α/β-tryptase heterotetramers with high biological activity consist of four non-covalently bound subunits, each monomer containing an active enzyme site, and oriented towards an inner face of a central pore to restrict access to serine protease inhibitors [70]. Heterotetramer formation follows two-steps proteolytic processing and stabilization by heparin proteoglycans. In a first step, tryptase monomers with much less activity form by autocatalytic intermolecular cleavage at acidic pH (5.5–6.5) and in the presence of heparin or dextran sulphate. On the other hand, tryptase tetramers form after the removal of the remaining precursor dipeptide by dipeptidyl peptidase I, which also requires heparin or dextran sulphate [71,72,73,74,75,76,77]. However, the importance of sequential autocatalytic then dipeptidyl peptidase I-mediated processing of pro alpha and beta to mature tryptases has been strongly challenged, as genetically modified mice lacking dipeptidyl peptidase I are able to produce active mast cell tryptase [78]. Additionally, recent studies have shown that monomeric pro-tryptases mature directly to active tetramers by cathepsins B and L in human mast cells [79,80]. Highly active heparin-bound tetrameric peptides are stored in MC granules and are released upon MC activation [71,72,73,74,75,76,77]. Heparin dissociation can lead to a rapid degradation of tryptase tetramers into enzymatic inactive monomers [77]. Alpha-tryptase also forms mature homotetramers [81]. Although the DNA homology of α- and β-tryptase genes is challenging for some molecular tests [82], α- and β-II tryptase show a number of amino acid differences in the catalytic domain, including differences in pro-peptide content affecting the potential for autoactivation, a key amino acid substitution in the substrate binding cleft key greatly reducing catalytic capability, and one of two N-linked glycosylation sites, potentially affecting stability and other physicochemical properties. Subsequently, mature α-tryptase has more restricted substrate specificity compared to β-tryptase and only negligible in vitro proteolytic activity [83,84].

## 3. Regulation of Tryptase Expression and Secretion

Human MCs are assumed to originate from hematopoietic stem cells of the bone marrow (BM), and further develop from multipotent MC progenitors under the influence of certain transcription factors and cytokines, with microphthalmia-associated transcription factor (MITF) and SCF playing a key role in MC differentiation [9,10,23,85,86,87,88,89]. Studies in murine MCs and MC progenitor cells have revealed that MC progenitors are able to leave the BM, migrate into various tissues and undergo differentiation to mature MCs with heterogeneous, tissue-specific phenotypical and functional properties [3,23,90,91,92,93,94,95,96,97,98]. SCF has also been described to be involved in the migration and survival of mature MCs and potentially also in MC activation [9,86,88,91,99,100]. Other cytokines, such as interleukin-4 (IL-4) and IL-6, were also demonstrated to regulate the proliferation, maturation and survival of MC progenitor cells and the function of mature MCs [11,13,101,102,103,104,105,106]. IL-3 also promotes the proliferation of MC progenitors, since these cells express IL-3 receptor (IL-3R). IL-6 has also been described to regulate proliferation in the early stages of MC development [13,105,106]. Mature tissue MCs lack IL-3 receptors [107,108]. IL-4 has been described to promote the survival and maturation of human MCs in later phases of MC development [11,103,104,109]. Another involvement of IL-4 in chymase expression has been reported [11,103]. Under certain conditions, IL-4 may also act as a negative regulator of the SCF-mediated differentiation of MCs [13,101,102,110,111].

Human MCs also show quite heterogeneous morphological properties during their development. While MC progenitors are non- or less granulated, and are considered to contain only low levels of MC-specific compounds, mature MCs are highly granulated and contain large amounts of preformed mediators [112,113]. In murine MCs, histamine synthesis has been described to be associated with granule maturation and has also been implicated in an autocrine regulation of early granule development [112,114]. Based on the expression of tryptase and chymase in their secretory granules, human MCs can be classified (divided) into MC_TC_ (MCs expressing both tryptase and chymase) and MC_T_ (MCs expressing only tryptase) types. MC_TC_ are primarily found in the skin, small intestinal submucosa and tonsils, while MC_T_ are typically localized in lung alveoli and in the small intestinal mucosa [115,116,117]. Additionally, in rodents, two distinct types of MCs with typical morphological features and different compositions of stored mediators were identified in various organs and tissue sites. Mucosal MCs (MMCs) are smaller and less granulated cells compared to connective tissue-type MCs (CTMCs). In comparison to MMCs, CTMCs contain abundant amounts of heparin [115,118,119].

Several other studies, particularly experiments on mouse or rat MCs, describe the transcriptional and epigenetic regulation of tryptase expression, as well as granule preformation and secretion, which is influenced by different extracellular factors, including cytokines and tissue environment, but also several intracellular molecules and processes. Early studies reported that the presence of proteoglycans (heparin and chondroitin sulphate E) and their interactions with tryptase play an important role in tryptase activity, due to enzyme stabilization and prolonged tryptase expression [120]. SCF is well known to augment tryptase expression in MCs in various species [114,121]. The two transcription factors Gata1 and Gata2 were reported to play a role in tryptase expression, since a reduction in the Gata transcription factors led to a decreased tryptase gene (*Tpsb2* and *Tpsg1*) expression in murine bone marrow MCs (BMMCs) [122,123]. Growth factor independent 1 (Gfi1) and its paralogue Gfi1b both play a crucial role in hematopoietic growth and differentiation. Experiments with these two transcription factors in murine BMMCs indicated that the one-sided expression of Gfi1/Gfi1b suppressed the induction of tryptase [124]. MITF has also been demonstrated to play a role in regulating the tryptase gene transcription in human MCs. Specifically, experiments with the human MC line HMC-1 showed an elevated tryptase production after the overexpression of the MITF-A isoform, and a lower tryptase expression and enzymatic activity by introducing an MITF-specific siRNA [125]. Research on the impact of copper transport protein Ctr2 on MCs revealed that an absence of the protein resulted in profound effects on the MC proteoglycans and MC maturation, as well as in the increased mRNA expression, storage and enzymatic activity of tryptase. Ctr2 knockout mice (*Ctr2*−/−) BMMCs showed an increased degree of maturation, contained more heparin, but less chondroitin sulfate proteoglycans and decreased levels of histamine. Additionally, an augmented chymase and tryptase activity and in vivo expression, and an upregulation of Mitf expression in the absence of Ctr2, were displayed [126]. Tryptase itself induces MC accumulation, as shown by Liu et al., who described that in the presence of a synthetic secretagogue of mast cell degranulation compound 48/80, tryptase contributed to induce MC accumulation in mice. Additionally, an inhibitory effect of tryptase inhibitors and PAR-2 antagonists on tryptase-induced MC accumulation has been reported [127]. Interestingly, two studies of Melo et al. demonstrated that tryptase could also be found in the nucleus, where it has the ability to proteolytically cleave core histone N-terminal tails and catalyze histone clipping (histones H2 and H3). Therefore, tryptase might also play an important role in MC epigenetic regulatory mechanisms through effects on the chromatin landscape and histone modification [128,129]. Grujic et al. reported a dynamic electrostatic relationship and balance between oppositely charged compounds and the regulation of MC granule composition [130].

## 4. Clinical Significance of Elevated Serum Tryptase Levels

A basal serum tryptase (BST) level reflects the constant tryptase (pro-α- and pro-β-tryptase) release from mature tissue MCs. An event-related β-tryptase release upon degranulation increases this baseline serum tryptase level and represents a marker of MC activation [69,131,132]. The measurement of human secreted total tryptase (pro and mature α- and β-tryptase, encoded by *TPSB2* and *TPSAB1*) is established as a routine laboratory parameter in clinical practice in the fields of allergy, immunology and hematology [68,69,131,132,133,134,135,136,137]. The majority of laboratories use a commercially available fluorescent-based immunoassay. 

The basal serum tryptase (BST) level in the human serum of healthy individuals is rather stable and averages at about 5 ng/mL in the majority of individuals. Based on the upper 95th percentile value determined in 126 healthy individuals aged 12–61 years, the upper reference cut-off was set to 11.4 ng/mL [69,131,138,139]. However, in other studies, the range of serum tryptase levels measured in heathy controls ranged between 0 and about 30 ng/mL with a proposed reference range of 0–15 ng/mL [133,134,135]. Further studies demonstrated (slightly) elevated BST levels with a prevalence of 4 to 6% in the Western general population, but the underlying reasons, biological mechanisms and clinical relevance remain rather unclear, although a genetic component has already been suspected [82,139,140,141,142]. Only a few inherited single-gene disorders have yet been identified to be associated with elevated BST levels, such as autosomal recessive Gaucher’s disease (GD) and GATA2 haploinsufficiency [143,144].

Acquired elevated serum tryptase levels can be detected in several clinical conditions and diseases. A markedly elevated serum tryptase level >15 ng/mL is a reliable biomarker for a subset of patients with myeloid neoplasms (AML—acute myeloid leukemia, CML—chronic myeloid leukemia, MPN—myeloproliferative neoplasms, MDS—myelodysplastic syndrome, CMML—chronic myelomonocytic leukemia and CEL—chronic eosinophilic leukemia) and MC diseases (e.g., systemic mastocytosis) and a helpful screening parameter in these hematologic disorders [131,133,134,145]. In addition, changes in BST over time may be indicative of disease progression or a response to therapy in these disorders [131,133,134,145]. In lymphoid neoplasms, including lymphoma and multiple myeloma, tryptase values are usually within the normal range [134]. In chronic myeloid leukemia (CML), basophilia is an independent risk factor and well-established prognostic variable in *BCR/ABL1*+ CML, and included in appropriate risk scores [146,147,148]. At the time of diagnosis, elevated serum tryptase levels can be detected in about 30–35% of CML patients, due to the increased production and secretion by immature CML basophils [134,149,150]. Elevated serum tryptase levels (>15 ng/mL) were shown to be more common in advanced CML than in chronic phase CML, and, compared to patients with normal tryptase levels, to be associated with significantly higher progression and event rates, and a lower response rate under tyrosine kinase inhibitor treatment [149]. Therefore, an elevated serum tryptase level at diagnosis is a significant prognostic biomarker in CML, and has been discussed as an interesting marker to be included in prognostic scoring systems [133,149].

Substantially elevated serum tryptase levels (>15 ng/mL) are also detectable in about 30–40% of patients with acute myeloid leukemia (AML), especially in AML FAB groups M0, M2, M3 and M4eo (M4eo with the highest tryptase concentrations), but not in patients with acute lymphoid leukemia (ALL) [133,134,151]. Sperr et al. demonstrated that tryptase is expressed and released by immature AML neoplastic cells (myeloblasts). They also described a correlation between serum tryptase levels and treatment response, since treatment induction resulted in a decrease in tryptase and the majority of patients achieved normal tryptase values at the time of complete remission. Hence, serum tryptase also represents a disease-related marker in diagnosis (reflects the total burden of leukemic cells) and disease monitoring in a subset of patients with AML [151,152]. However, acute myeloid leukemia with mast cell lineage involvement (SM-AML) is an exception in this context, as the resistant SM component of the disease prevents a substantial serum tryptase decrease during therapy [133].

Serum tryptase levels >15 ng/mL are also frequently detected in a subset of patients with other myeloid neoplasms, such as MDS (including MDS/MPN overlap and CMML), MPN and CEL, but are of less diagnostic or prognostic importance herein [133,134,153]. The higher tryptase amounts are considered to derive from basophils, mast cells or blasts, depending on the type of disease and on the presence of concomitant systemic mastocytosis [133,153]. Tryptase should always be measured in patients with AML and CML and in all cases with suspected mastocytosis. In MDS and MPN, tryptase measurements are only recommended at the time of diagnosis, unless massive basophilia or eosinophilia, a mastocytosis or a very high initial tryptase level was detected at diagnosis [133]. In patients with suspected MC disease, serum tryptase is a robust screening parameter [33,34,131,133,137]. While the majority of cutaneous mastocytosis (CM) patients were shown to have tryptase concentrations <15 ng/mL, the majority of patients (>90%) with systemic mastocytosis (SM) exhibit an elevated serum enzyme level >15 ng/mL [134]. In particular, a markedly elevated BST level of >20 ng/mL is a minor diagnostic criterion for systemic mastocytosis (SM) [33,34,136]. Depending on the subtype of SM and the MC burden, serum tryptase concentrations vary among patients. The prognostic value and clinical usage of BST in MC diseases will be discussed in the following paragraphs.

In contrast to BST, serum tryptase after MC activation is highly dynamic. In severe allergic reactions, anaphylaxis and systemic hypersensitivity reactions, a marked increase in the serum tryptase level above the individual´s baseline is an indicator and a clinical biomarker of substantial systemic MC activation [68,131,132,137,138,154,155]. In otherwise healthy individuals with normal baseline tryptase, in whom severe MC activation and anaphylactic shock are recorded, serum tryptase levels may peak up to 100 ng/mL 0.5–1.5 h after anaphylaxis onset [131]. Importantly, tryptase measurement after mediator-related MC degranulation has to be determined in blood samples within 3–4 h after the resolution of symptoms, due to a relatively short half-life of tryptase [131,137,155,156]. Because of different individual tryptase baselines, an event-related tryptase value of at least a 20% increase over the individual´s baseline + 2 ng/mL (1.2 × BST + 2 ng/mL) is used for the definition of an MC activation in MC activation syndromes (MCAS) [138,154,155].

## 5. Tryptase in MC Diseases

Mastocytosis is a term used for a group of hematopoietic neoplasms characterized by a pathological increase in and accumulation of clonal MCs in the skin (cutaneous mastocytosis: CM) or in one or more extracutaneous organ systems (gastrointestinal tract, BM, liver, spleen, lymph nodes and others), named systemic mastocytosis (SM). It can occur at any age and with variable mediator-related symptoms and heterogeneous clinical courses, from mild, sometimes self-limiting forms to severe, advanced forms with a shorter overall survival [14,17,33,34,157,158,159]. According to the World Health Organization (WHO) classification, mastocytosis is divided into CM, SM and mast cell sarcoma (MCS) [33,136,160]. SM is classified into indolent SM (ISM), smoldering SM (SSM), SM with an associated hematologic neoplasm (SM-AHN), aggressive SM (ASM) and MC leukemia (MCL) [33,34,136,160]. Patients with CM and ISM have the best prognosis with a normal or near normal life expectancy, while patients with SM-AHN, ASM, MCL and MCS have an unfavorable prognosis often with rapid disease progression [33,136,157,158,159,160,161,162,163].

CM is particularly seen in (early) childhood and is divided into three major forms. Clinical features of maculopapular CM (=urticarial pigmentosa) are maculopapular pigmented and often reddish skin lesions [164,165]. Other forms are diffuse CM (DCM) and mastocytoma of skin. Patients with DCM may present with more severe symptoms and sometimes even blistering of the skin. SM is usually diagnosed in adults and is often accompanied by skin lesions, especially in patients with ISM and SSM. Symptoms in SM can be grouped into constitutional symptoms (e.g., fatigue and weight loss), symptoms related to skin (e.g., pruritus, urticaria and flushing), systemic mediator-related symptoms (e.g., headache, hypotension, tachycardia, respiratory and gastrointestinal symptoms and abdominal pain), musculoskeletal symptoms (e.g., bone pain due to osteoporosis or osteopenia and myalgia) and symptoms caused by mast cell infiltration with subsequent organ damage seen in advanced SM (cytopenia-related symptoms, pain related to organ destruction, ascites and splenomegaly-related symptoms) [136].

The diagnosis of mastocytosis is established by histological examination of the affected organ system(s) where MC infiltration can be documented, e.g., in the skin and/or the BM by immunohistochemical stains. In addition, depending on the type of SM, laboratory parameters show typical results, such as abnormal cell surface marker expression on MCs, elevated serum tryptase level, elevated alkaline phosphatase activity, cytopenia, or leukocytosis. In SM, molecular studies may reveal the presence of a *KIT* mutation (usually D816V) in neoplastic cells and sometimes chromosome abnormalities or additional mutations in other target genes (apart from *KIT*), especially in SM-AHN. All patients with SM also undergo routine clinical staging, in order to document or exclude B- (=Borderline Benign) and C-findings (=Consider Cytoreduction or Chemotherapy). The diagnosis of SM is based on major and minor diagnostic criteria defined in the WHO classification. The major SM criterion is the multifocal clustering of MCs (≥15 cells per aggregate) in the BM or in another extra-cutaneous organ system. Minor criteria relate to the abnormal morphology of MCs, abnormal (aberrant) expression of CD2 and/or CD25 in MCs, an activating mutation in codon 816 of *KIT* and an elevated tryptase level exceeding 20 ng/mL. When at least one major and one minor or at least three minor criteria are fulfilled, the diagnosis of SM can be established [14,17,33,34,136,160]. In the majority of all patients with SM, a somatic amino acid substitution of valine for aspartic acid in the catalytic domain of *KIT* (p.Asp816Val or p.D816V) can be detected, which leads to a constitutive activation of the receptor. Other activating, pathological variants in *KIT* have also been described—these are less frequently detected in patients with SM but are more frequently detected in patients with childhood CM [166,167,168].

For diagnosis and risk assessment of patients with SM, so-called B-findings, which indicate a huge burden of neoplastic (mast) cells and signs of multilineage involvement without organ damage, and so-called C-findings, which indicate organ damage produced by a massive infiltration with neoplastic MCs, have been established. Patients with 2 or 3 B-findings but no C-finding are suffering from smoldering SM, whereas patients in whom one or more C-findings are detected are suffering from aggressive SM (ASM) or another type of an advanced SM. The management and treatment of patients with mastocytosis is based on the type of disease (variant of CM or SM), SM-related organ damage (C-findings) and mediator-related symptoms [33,34,136,145,160,169,170]. So-called “triggers” (all kinds of allergens, such as drugs, insect venom and foods) and individual factors that may induce MC degranulation and severe anaphylactic reactions, a common problem in mastocytosis patients, have to be identified and avoided whenever possible [171].

An elevated serum tryptase level is an important diagnostic parameter and clinical biomarker in MC disorders. An elevated persistent BST level >20 ng/mL is a minor SM criterion according to the WHO classification. However, this criterion is only valid in the absence of an AHN because the AHN component of the disease may contribute to the increased BST [33,136,145,160]. It is also worth noting that an elevated BST level (even if very high) is not a marker of MC activation. Rather severe MC activation and MCAS are more frequently seen in those SM patients who have a lower basal tryptase level, and only an acute event-related increase in tryptase above the individual´s baseline (following the 20% + 2 equation) qualifies as a biomarker of systemic MC activation and thus as a criterion of MCAS. Very high BST levels are associated with less favorable prognosis and represent a B-finding in SM (>200 ng/mL + >30% infiltration of the BM biopsy by MCs) [33,34,135,138,145,155].

## 6. Genetic Background of Tryptase

In the late 1980s and early 1990s, the first studies described the genetic structures and functions of the human tryptase genes, and mapped sequences to a gene cluster on human chromosome 16 by PCR analysis of DNA from human/hamster somatic cell hybrids, as well as bacterial artificial chromosome (BAC) analysis and fluorescence in situ hybridization (FISH). Further research revealed multiple DNA sequences encoding tryptase with close localization and high similarity [172,173,174,175]. The currently known tryptase isoforms are α-I tryptase, α-II tryptase, β-I tryptase, β-II tryptase, β-III tryptase, γ-tryptase and δ-tryptase. Five genes were identified encoding for these tryptase isoforms, all located within the gene-rich and highly repetitive genomic region 16p13.3 on the short arm of chromosome 16. *TPSG1* (encoding γ-tryptase), *TPSB2* (encoding β-tryptase), *TPSAB1* encoding (α- and β-tryptase) and *TPSD1* (encoding δ-tryptase) are localized paralogous genes within a gene cluster. Investigation of the tryptase locus revealed that primary secreted and biologically relevant soluble tryptase, expressed primarily by MCs and basophils, derive only from the *TPSB2* gene, encoding only β-tryptase (β-II or β-III tryptase isoforms), and from the *TPSAB1* gene, encoding either α- or β-tryptase (α-(α-Ι) or β-I tryptase isoforms). Both genes are arranged in tandem, consisting of 6 exons and have very strong homology in their sequences. Beta-tryptase isoforms are 98–99% identical in their DNA sequence, but there are multiple amino acid differences in the catalytic domains between β-III and both β-I and β-II. In addition to an 11-base-pair deletion in intron 4, the *TPSAB1* α-allele differs from the β-II allele in several exchanges of single bases in multiple exons, including a number of amino acids in the catalytic domain that critically affect substrate specificity and proteolytic activity [66,67,172,173,174,175,176,177]. Despite these functionally highly relevant differences in key amino acids, the overall high homology (~97%) of the DNA sequence of the α-tryptase gene with β-tryptase genes demands specific molecular techniques for tryptase genotyping and limits the validity of SNP-array-based genomewide association studies (GWAS) for this locus [82].

Further genotyping and isoform expression analysis of *TPSAB1* and *TPSB2* tryptase genes revealed the three most common genotypes, considered as canonical: 0α:4β (also named α-tryptase deficiency), 1α:3β and 2α:2β. Individuals with a 2α:2β gene ratio carry the two α-tryptase copies on opposite alleles (in trans), one being inherited from each parent. Genotyped DNA from 274 individuals found α-tryptase deficiency in 29% of individuals and mixed genotypes in 21 (2α:2β) and 50% (1α:3β) [178]. Studying the prevalence of α-tryptase deficiency showed that it is quite common and differently distributed in ethnical groups (most common in Caucasians, less in Asians), while no individuals lacking β-tryptase have been reported to date. Although the reason for the differential biological stability of α- and β-tryptase is not entirely clear, these observations are in line with the notion that α-tryptase exhibits no or minimal (biological) protease activity contrasting the multifunctional properties of active β-tryptase. The involvement of β-tryptase in allergic disorders has further suggested that inherited differences in α-β genotypes might affect disease susceptibility, severity and response to therapy [178,179].

*TPSD1* is the δ-tryptase locus encoding Mast-Cell-Protease 7-LIKE (MCP7-like-I, also δ-I) and MCP7-like-II (δ-II). The sequences of δ-I and δ-II tryptase differ only at two nucleotides and one amino acid. The structural similarity of δ- and α/β-tryptase is approximately 80%, but the protein is truncated and largely catalytically inactive, due to a premature translation termination codon and less substrate specificity. δ-tryptases were found in the MCs of the large intestine, lungs and heart [172,180,181]. *TPSG1* (also TMT—transmembrane tryptase) encoding γ-Tryptase I and II has approximately 48% similarity to α/β-tryptase. Gamma-(γ)-tryptases are transmembrane proteins bound to the plasma membrane or to the membrane of secretory granules. They have a relatively low expression rate and quite different substrate specificities [177,182,183].

A number of techniques have been applied to detect and delineate the expression of common forms and variants of the human tryptase genes, including common sequencing or PCR technologies at the DNA level. Previous technologies with high resolution melt curve analysis, Sanger sequencing and Southern blotting after restriction enzyme digestion provided relative quantitation of *TPSAB1* and *TPSB2* by chromatogram peak height or banding sizes. However, there were several limitations in base mapping and interpretation due to the high homologies of the tryptase genes. Next-generation sequencing (NGS) may overcome some of these limitations using bioinformatics applications [82]. Nevertheless, restrictions remain because accuracy in copy number analysis is highly dependent on primer design, read depth and read quality. To date, droplet digital PCR (ddPCR) is the most common, robust and reliable method in the detection of copy number variations in the *TPSAB1* gene, and allows the determination of the most allelic genotypes [82,184,185]. *TPSAB1* CNV detection by ddPCR allows quantification of the allelic α- and β-tryptase copies with primers and fluorophore-labelled probes targeting specifically α- and β-tryptase sequences. DNA digestion with Bam HI high-fidelity restriction enzyme is required to achieve separated tandem gene copies for an accurate and optimal copy number measurement. Within the assay, the parallel amplification of a reference gene with a known copy number status (usually two copies, such as *AP3B1*) labelled with another fluorophore is necessary for comparing relative concentrations of the target *TPSAB1* and the reference gene. This allows a conclusion about the *TPSAB1* copy number state (α-tryptase and β-tryptase copies) [82,186].

## 7. Hereditary Alpha Tryptasemia

In 2014, Lyons J.J. et al. reported severe, unique, allergic phenotypes and elevated BST levels without any diagnosable evidence of an underlying causative disease in several families. Clinical presentations and symptoms, such as urticaria, flushing, cramping abdominal pain, diarrhea and a history of anaphylaxis (food-mediated, insect stings and idiopathic), were reported to be chronic or episodic, sometimes without a known trigger or induced by certain triggers, such as exercise, vibration, stress, food, heat or physical trauma. Because of the similarity to MC mediator-related symptoms in SM affecting multiple organ systems (e.g., headache, bone pain, nausea, itching, diarrhea, hypotension and anaphylaxis) and an elevated BST, an underlying MC disease was suspected. However, further diagnostic investigations revealed that many of the affected patients examined had no evidence of a clonal MC disease or an MC activation syndrome (MCAS). Interestingly, the elevated BST levels (mean BST 21.6 ± 1.4 ng/mL), named tryptasemia, and the clinical symptomatology were found to cluster within families. Therefore, an autosomal dominant inherited manner was assumed, but a specific cause or an associated biological mechanism remained unclear until 2016 [141].

In 2016, Lyons J.J. et al. first described hereditary α-tryptasemia (HαT), a biochemical and genetic trait that is associated with elevated serum tryptase levels. It is caused by germline copy-number gains (duplications, triplications or more copies) on one single allele in the *TPSAB1* gene encoding α-tryptase (Figure 1) [82]. The authors investigated 35 families (137 individuals) with a frequent occurrence of various clinical manifestations, including cutaneous flushing, pruritus, urticaria, autonomic dysfunction (including postural orthostatic tachycardia syndrome and POTS), functional gastrointestinal symptoms (irritable bowel syndrome, dysmotility and chronic gastroesophageal reflux), pain (chronic arthralgia, headache and body pain), complaints of sleep disruption, systemic reaction to stinging insects (e.g., Hymenoptera) and connective tissue abnormalities (joint hypermobility, congenital skeletal abnormalities and retained primary dentition) [82]. A high normal to slightly elevated BST level (>8 ng/mL) was found in all individuals tested, and some of these individuals had clearly elevated BST (>20 ng/mL). Exome and genome sequencing and linkage and segregation analysis did not reveal shared rare or frequent variants in the affected family members, but identified a single deviation in the human tryptase locus on chromosome 16p13.3 [82]. Lyons and colleagues performed additional analysis of the *TPSAB1* gene, encoding for tryptase and located within this region, with an adapted and validated digital droplet PCR (ddPCR) assay, which allowed accurate differentiation, detection and direct quantification of copy numbers of the α- and β-tryptase sequences [82]. *TPSAB1* analysis yielded in the identification of increased α-tryptase copy numbers (αα- two copies or ααα- three copies on one single allele) in individuals with elevated BST levels (Figure 1) [82]. An overall of 96 of the 137 individuals of the 35 families were identified with inherited mono-allelic extra alpha copies in the *TPSAB1* gene, thus showing an autosomal dominant pattern [82]. Not all of these 96 HαT individuals showed a clinical phenotype, suggesting additional genetic variants, epigenetic regulations and co-morbidities being involved [82]. Especially co-morbidities, such as IgE-dependent or IgE-independent allergies, mastocytosis or immunological disorders may determine the clinical picture in HαT carriers. In patients with SM and HαT, the serum tryptase level may be particularly elevated. A gene dose effect was also assumed, since a higher number of α-tryptase copies could be associated with higher tryptase values and a higher prevalence of severe clinical symptoms [82]. This gene–dose effect was later also found by Sabato et al., who reported that germline *TPSAB1* α-quintuplications (five α-copies on one allele) in a Belgian family were associated with significantly higher BST levels compared to individuals with duplications or triplications in the *TPSAB1* gene [140].

The examination of additional cohorts of individuals with available BST levels confirmed a common occurrence of HαT in the general population. The overall estimated prevalence of HαT may range between 3 and 8% in the Western world [139,142]. The prevalence of HαT with additional *TPSAB1* α-tryptase copies was found in 8.2% (8/98) of the study cohort and in 7.2% (9/125) of the control cohort, determined by exome/genome sequencing data and *TPSAB1* CNV ddPCR [82]. In another recent study of Robey et al., HαT was identified in 5% (22/423) of individuals in an unselected British birth cohort with a median BST level of 11.7 ng/mL [187]. In this study, a similar patient number of 5% (204/4283) of a regional immunological laboratory from the UK, screened for serum tryptase due to their clinical symptoms, had an elevated BST level equal to or greater than 8 ng/mL (median of 12.9 ng/mL). The most frequent clinical symptoms of a further 70 patients diagnosed with HαT were urticaria or angioedema, skin flushing and abdominal symptoms, occurring with diverse severity and variable penetrance within two families (1 with *TPSAB1* α-quintuplication and 1 with *TPSAB1* α-triplication). The observed differences in clinical presentation could be influenced by co-morbidities or additional yet unknown genetic variations. Median BST level of 79% of patients with *TPSAB1* α-allele duplication was 15.2 ng/mL, while in the 21% with α-allele triplication or higher copy number measures, BST was 22.0 ng/mL, resulting in an overall BST median of 17.0 ng/mL. Members of the *TPSAB1* α-quintuplication family had a median BST level of 40.0 ng/mL [187].

In 2020, Lyons J.J. et al. described a significantly higher prevalence of HαT in patients with idiopathic anaphylaxis (17–8/47) compared to a HαT prevalence of 5.6% in healthy individuals (7/125) and in controls with nonatopic disease (5.3%; 21/398). They also found that increased *TPSAB1* α-tryptase copy number gains were associated with an increased risk for severe Hymenoptera venom-triggered anaphylaxis [188]. In summary, HαT is represented with a frequency of about 3–8% in the general population, and individuals with *TPSAB1* alpha extra copies have been described with elevated BST levels, but may also present with BST levels of >8.0 ng/mL but less than 11.4 ng/mL. In particular, additional *TPSAB1* alpha copy numbers resulted in elevated BST levels of 15 ± 5 ng/mL in those with one extra gene copy number, 24 ± 6 ng/mL in those with two extra gene copies and 37 ± 14 ng/mL in those with four extra copies [189]. Sometimes, BST levels can even exceed 100 ng/mL in patients with multiple extra tryptase gene copy numbers, which is clinically relevant in mastocytosis contexts. The term Hereditary alpha-Tryptasemia Syndrome (HαTS) has been used for HαT individuals with clinical manifestations and multisystem complaints of varying severity as described before. However, the plethora of potential symptoms makes it challenging to differentiate between HαT-associated and other symptoms [82,187,189].

## 8. HαT—In Vitro Experiments and Possible Mechanisms of MC (Hyper)Activation

In vitro cultures of primary MCs derived from CD34+ blood cells of individuals with HαT showed increased *TPSAB1* and *TPSB2* mRNA levels and more spontaneous tryptase secretion compared to cultures of control without HαT. In contrast, no differences in cell growth and morphology, intracellular tryptase protein expression or IgE-mediated degranulation activity were observed. The increased translation and constitutive secretion of α-pro-tryptase in primary MC cultures were most likely the main causes of elevated BST levels taking the gene dosage effect into consideration [82]. However, the biological mechanism behind HαT and how elevated BST levels contribute to the associated multisystem disorder remained unclear at that point, as the biological effect of α-tryptase homotetramers has been questioned. Mature tryptase tetramers stored in secretory granules of MCs are formed from monomeric pro-tryptase precursors (α and/or β isoforms) in the presence of stabilizing heparin and cathepsins. Previous research on tryptase proteins showed that while β-tryptase homotetramers are active proteases, α-tryptase homotetramers exhibit negligible or no proteolytic activity [67,68,70,71,76,77,81,83,84,177,190].

A subsequent study by Le et al., investigating tryptase protein complexes in MC secretory granules, reported the natural formation of 2α- and 2β-tryptase heterotetramers. These heterotetramers are particularly increased in individuals with HαT and correlate with higher symptom severity [191]. Moreover, they were described to differ from α/β-tryptase homotetramers by enhanced proteolytic enzyme activity and functional properties, such as higher stability. Because these heterotetramers might also influence potential targets, authors focused on the in-vitro influence on the two proteins EMR2 (EGF-like module-containing mucin-like hormone receptor-like 2) and PAR2 (protease-activated receptor-2). EMR2, based on its involvement in hereditary vibratory urticaria, and PAR2, based on its activation by tryptase and stimulatory effects on neuronal, smooth muscle, endothelial, and inflammatory cells [191]. EMR2 represents a protein involved in hereditary severe vibratory urticaria, caused by a germline missense mutation in *ADGRE2* (p.C492Y). Mutation dependent altered EMR2α:EMR2β subunit binding leads to easier mechanical force (vibration) induced cleavage of the subunits and prolonged EMR2β dependent MC degranulation [192]. Incubation of skin MCs only with 2α-2β-tryptase heterotetramers compared to β-homotetramer exposure led to vibration-triggered MC degranulation by proteolytic cleavage of the EMR2 subunits [191]. Because urticaria is also a common clinical finding in HαT patients, they additionally performed experiments with 56 volunteers (35 healthy and 21 HαT individuals) to investigate response to vibratory cutaneous challenge. Only individuals with a HαT genotype showed a vibration-induced cutaneous effect [191].

The activation of PAR2 (synonym: F2RL1—F2R-Like-Trypsin-Receptor-1), a cell surface transmembrane receptor expressed by many cell types (e.g., endothelia, neurons, smooth muscle cells and several immune cells), by tryptase may lead to an increase in the expression and release of certain mediators and cytokines, such as TNF-α and IL-6. PAR2 is also a stimulator of diverse physiological processes (vascular smooth muscle relaxation, dilate blood vessels, increase blood flow and lower blood pressure) and a modulator of inflammatory response (during infection and in innate and adaptive immunity) [193,194,195]. In vitro experiments on Jurkat cells showed that heparin-stabilized 2α-2β-tryptase heterotetramers, but not β-homotetramers, lead to a dose-dependent PAR2 activation [191]. Authors suggested that these heterotetramer abilities might contribute to some of the symptoms described in HαT carriers by the activation of MCs and other cell types and might also be involved in disease severity [191]. To further investigate the mechanisms underlying the vascular effects of HαT on affected patients, transwell in vitro experiments were performed. These experiments revealed that, in contrast to α-β heterotetramers, only mature tryptase has the ability to promote the PAR2-dependent transmigration of human umbilical endothelial vein cells in vitro. An attractive hypothesis is that this enhanced vascular permeability via PAR2 activation plays a role in severe anaphylactic reactions in patients with HαT [188].

## 9. HαT in Mast Cell Diseases

Many symptoms attributed to MC activation, such as urticarial or anaphylaxis, were also reported in individuals with HαT [82,188,189]. Clonal MC diseases may also present with a wide variety of multiorgan symptoms, some of them overlapping with symptoms recorded in HαT carriers. These symptoms include pruritus, hypotension, flushing, bone pain and anaphylaxis. In many cases, an IgE-dependent allergy to Hymenoptera venoms, food allergens or other allergens is found. Patients with SM who are HαT carriers and have a concomitant IgE-dependent allergy (e.g., against bee or wasp venom) may be at a very high risk to develop severe life-threatening anaphylactic episodes. Furthermore, as an elevated BST to >20 ng/mL is a minor criterion for SM, it may be of interest to determine from genetic testing of *TPSAB1* whether the patient has HαT and whether HαT may contribute to elevated BST levels. For example, Sabato et al. reported a Belgian family with inherited *TPSAB1* α-quintuplications (five mono-allelic α-copies) and correspondingly high BST levels in 7 out of 15 family members. Only four of the affected individuals showed gastrointestinal symptoms, including severe abdominal cramping and diarrhea. In a 22-year-old male with mild hepatosplenomegaly, further investigation revealed the presence of a somatic *KIT* D816V mutation in BM cells, although the patient did not meet criteria for SM. The authors assumed the possibility of a pre-disease phase, and indicated a relationship of increased *TPSAB1* copy numbers and gene–dosage in the development and clinical manifestations of clonal MC diseases, such as mastocytosis [140]. The importance of distinguishing between clonal MC disease and HαT, and the diagnostic impact of genetic testing of *TPSAB1* copies in symptomatic patients before an invasive BM biopsy is considered, was also shown in three individuals of a clinical case report by Carrigan C. et al. [196].

In our recent study, we evaluated *TPSAB1* copy numbers and thus the HαT status in a larger cohort of mastocytosis patients (180 patients: 16 CM, 118 ISM, 10 SSM, 9 ASM, 3 MCL, 19 SM-AHN and 5 MIS—mastocytosis in the skin) [186]. The results of this study revealed a high frequency of HαT carriers (17.2%; 31/180) among the mastocytosis patients tested. This could also be confirmed in 61 additional mastocytosis patients of an independent validation cohort (HαT found in 21.3%) [186]. Interestingly, the highest prevalence of HαT was found in patients with indolent SM (18.8%; 24/128). Mastocytosis patients with HαT exhibited higher tryptase levels (independent of the MC burden), a significantly lower *KIT* D816V allele burden and more severe mediator-related symptoms than patients without HαT [186]. Thus, the presence of HαT might influence the prognostic value of markers for disease burden, such as BST and *KIT* D816V allele burden in SM [135,197,198]. Clinically, it is of particular importance that a strong association between HαT and Hymenoptera venom hypersensitivity reactions (30%) as well as severe cardiovascular mediator-related symptoms (anaphylaxis and hypotension; 35.5%) was found in mastocytosis [186]. The control group’s prevalence of HαT was 4.4% in 180 anonymized sex-matched controls, and 3.9 to 5.0% in 720 patients diagnosed with other myeloid neoplasms (180 AML, 180 MDS, MDS/MPN, 180 MPN and 180 CML), thus comparable to frequencies in the general Caucasian population as already published by other studies [186]. Our results were also confirmed by an independent study conducted by Lyons J.J. et al. [188]. This study focused more on the prevalence and impact of HαT in anaphylaxis, and showed a high frequency of HαT carriers among patients with grade IV (Hymenoptera) venom anaphylaxis and (severe) idiopathic anaphylaxis when compared to control groups lacking HαT [188]. In a cohort of mastocytosis patients, they also reported a high prevalence of HαT in SM (12.2%—10/82) and, similar to our study, an increased risk for severe systemic anaphylaxis within this group of HαT-positive SM patients. The observed frequency of HαT within two control groups (5.3–5.6%) was comparable to that observed in the Viennese cohort [188]. Based on these observations, HαT is considered a new biomarker to identify patients at risk of developing severe anaphylaxis in mastocytosis. Moreover, the determination of *TPSAB1* alpha extra copies is currently discussed as an important biomarker to be included in risk assessment models and in future diagnostic algorithms in patients with mastocytosis. Further studies are required to demonstrate how *TPSAB1* CNV-based risk prediction can be incorporated into clinical practice for individualized risk assessment and multiparametric prognostication models in mastocytosis and other diseases. It remains to be shown whether mastocytosis patients with HαT benefit from a more intense prophylactic management, or if *TPSAB1* gain could even present a predictive biomarker for specific treatment approaches.

In addition, an adjustment of the diagnostic BST threshold levels (determining SM criteria and B-findings) in mastocytosis in light of HαT-mediated increases in BST might be considered in the future [186]. Giannetti et al. recently described unique BM MC phenotypes and discrete histopathological changes in patients with MC activation symptoms and elevated BST, some of them diagnosed with HαT [199]. The MCs in these patients were atypical, larger, hypogranular and showed abnormal paratrabecular and/or perivascular localization, as well as associated eosinophilia. From their data, they assumed that HαT might be associated with intrinsic MC abnormalities that may contribute to or predispose for the development of SM and predispose to MC activation and MCAS at the same time [199]. To elucidate the mechanistic link between the high prevalence of HαT and mastocytosis requires forthcoming studies. Extra tryptase (variants) might stimulate mitogenic activity not only on cells of the BM microenvironment but also on hematopoietic stem cells, triggering the development of SM. To identify autocrine/paracrine effects of tryptase on these cells, further studies need to include in vitro experiments with *KIT* D816V+ mast cell lines, cultured cord-blood cell-derived mast cells, induced pluripotent stem cells derived from *KIT* D816V+ HαT+ mastocytosis patients and primary patient samples as well in vivo experiments with murine xenotransplant models.

## 10. Therapeutic Approaches in HαT-Mediated (Co-Triggered) Pathologies

First, it is of the utmost importance to state that the HαT carrier status or HαT itself is neither disease initiating nor an indication for the treatment of patients. Thus, in individuals with HαT that do not suffer from an allergy, MCAS or another disease producing MC activation, no treatment is required. In patients with symptoms of MC activation, treatment strategies to manage MC activation are applied according to general and local recommendations for treatment of the underlying disorders (allergy/anaphylaxis or MCAS or mastocytosis with symptoms CM_SY_/SM_SY_). In the long-term management of (severe) allergy and anaphylaxis, the European Competence Network on Mastocytosis (ECNM) and the World Allergy Organization (WAO 2020 guidelines) recommend the identification of triggers, the avoidance of any triggering factors and the prevention of recurrence by drug therapy or allergen immunotherapy and/or desensitization, as well as a personalized anaphylaxis emergency action plan with the use of epinephrine in emergency situations (e.g., epi-pen) after detailed instructions [155,160,170,200,201]. Currently, no data are available on whether the presence of HαT influences the response of individuals to specific medications or immunotherapy. However, severe allergic reactions and anaphylaxis are frequently observed in allergic patients with HαT and especially in mastocytosis patients with HαT, and many of these patients may develop severe reactions or even MCAS. Therefore, a personalized anaphylaxis emergency action plan is of particular importance in these patients [170,186,200].

In addition to these general approaches, the first results of more specific treatment approaches have been described. The role of omalizumab, a monoclonal anti-IgE antibody that binds to free IgE and reduces the expression of FcεRI on MCs and basophils, was discussed in the management of selected individuals with HαT who suffer from severe, IgE-dependent, allergies or secondary (IgE-mediated) MCAS. A comprehensive retrospective multicenter study led in 2020 by Mendoza Alvarez et al. investigated treatment response to omalizumab in a small cohort of 13 HαT patients (eight with 3α:2β, two with 2α:3β and three with 4α:2β) [202]. The aims were to compare the patient´s symptoms before and after treatment, to find out which symptoms are mainly caused by HαT and are potentially associated with an increase in heterotetrameric tryptase in MCs. After a median time of eight weeks, 12 of 13 patients experienced an improvement in nearly half of the reported clinical symptoms, especially urticaria, nausea, flushing, fatigue and abdominal pain, although individual responses were quite variable [202]. In contrast, symptoms such as autonomic dysfunction, palpitations, diarrhea or joint hypermobility showed no improvement [202]. Recently, Maun HR et al. generated a noncompetitive inhibitory antibody that binds to human α- and β-tryptases, and potently inhibits active β-*tryptases* in vitro by the dissociation of active tetramers into inactive monomers. Furthermore, anti-tryptase reduced IgE-induced systemic anaphylaxis responses in a humanized mouse model providing a scientific rationale for testing anti-tryptase clinically [203]. The blocking of tryptase might also improve the quality of life in other MC-mediated diseases, including HαT and mastocytosis. Those patients might benefit from tryptase inhibition that can lead to a lower frequency and/or severity of tryptase-induced symptoms. Furthermore, disease course might be modified by abolishing tryptase stimulated autocrine/paracrine signaling loops between MC, cells of the BM microenvironment or mesenchymal cells. In principle, such antibodies may represent a therapeutic option in HαT patients, because these agents can also inhibit α/β-tryptase heterotetramers, as they bind not only β- but also α- and β-isoforms [203]. However, no clinical data of the effects of anti-tryptase antibodies on patients with HαT are available to date.

## 11. Concluding Remarks and Future Perspectives

The discovery of HαT has been crucial for our understanding of tryptase genetics and the mechanism of regulation of tryptase expression and secretion in health and disease. Clinically, HαT represents a highly relevant risk factor for severe mediator-induced symptoms, such as anaphylaxis, in patients with hypersensitivity disorders, including IgE-dependent allergies and patients with mastocytosis. Therefore, the *TPSAB1* status should be considered as a new predictive biomarker in these patients and should be included in routine diagnostic algorithms and future clinical studies. Furthermore, HαT is an important differential diagnosis for individuals presenting with a slightly elevated BST and no signs or symptoms indicative of SM or a hematologic non-MC neoplasm. Further studies are needed to elucidate if and how HαT directly contributes to the pathogenesis of MC activation and primary MC diseases. Finally, a detailed mechanistic understanding of HαT may contribute to the development of new specific therapeutic approaches for patients with MC disorders.

## Figures and Tables

**Figure 1 ijms-22-02458-f001:**
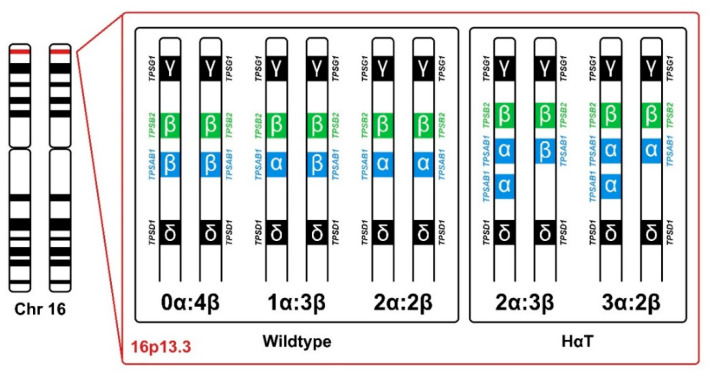
Canonical genotypes (wildtype) of α - and β-tryptase derived by *TPSAB1* and *TPSB2* copy number state, and (most common) genotypes of HαT (hereditary alpha tryptasemia) with additional *TPSAB1* α-copies.

## Data Availability

Not applicable.

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
