# Peer review of "Genetic Regulation of Tryptase Production and Clinical Impact: Hereditary Alpha Tryptasemia, Mastocytosis and Beyond"

_ijms, 2021, doi:10.3390/ijms22052458_

Round 1
Reviewer 1 Report
"Genetic regulation of tryptase production and clinical impact:hereditary alpha tryptasemia, mastocytosis and beyond"
This is a good review that summarizes the impact of tryptase production in
the mast cell accumulation and other clinical features of hereditarty
alpha-tryptasemia. Most key findings from the available literature on
herditary alpha tryptasemia and its link to mast cell accumulation, degranulation,
diagnosis and therapeutic targeting are included.
However, it will be more useful if the authors also include
their views, ideas and intriguing questions for future research in this area.
Major points
It will be very useful if a detailed description of methods for tryptase genotyping is included. For example, additional information on the details of droplet digital PCR and the method by which it allows the detection of TPSAB1 and TPSB2 allele-specific genotyping will be very helpful.
Mechanisms associated with mastocytosis and hereditary α-tryptasemia, the authors' thoughts, ideas, perspectives gained from previous research are valuable to provide more insight to the readers. Those studies that examined the influence of hereditary alpha tryptasemia on mast cell activation including those that tried to elucidate or hypothesize the molecular differences between the mutat KIT and WT KIT receptor and the impact on their localization, function, and susceptibility to phosphatases, etc are interesting determinants that need to be included. Specific details that could provide any insight on the mechanism by which gain in TPSABP1 copy number can lead to the mastocytosis and the clinical features of mastocytosis will be of added value. More details on how does mast cell tryptase serve as a potential target and additional descriptions about the potential mechanisms by which anti-tryptase antibody can block anaphylaxis/severe asthma in patients with mastocytosis could provide more interest to readers. What about patients with hereditary α-tryptasemia but failed to show any evidence of mastocytosis?
The scope of developing genetic biomarkers by identifying the gain in TPSAB1 copy number for anaphylaxis risk prediction in patients with mastocytosis should be elaborately discussed.
Minor points
page 7 line 306: by immunohistochemically stains.
Author Response
Reviewer 1:
"Genetic regulation of tryptase production and clinical impact: hereditary alpha tryptasemia, mastocytosis and beyond". This is a good review that summarizes the impact of tryptase production in the mast cell accumulation and other clinical features of hereditary alpha-tryptasemia. Most key findings from the available literature on hereditary alpha tryptasemia and its link to mast cell accumulation, degranulation, diagnosis and therapeutic targeting are included. However, it will be more useful if the authors also include their views, ideas and intriguing questions for future research in this area.
Major points
It will be very useful if a detailed description of methods for tryptase genotyping is included. For example, additional information on the details of droplet digital PCR and the method by which it allows the detection of TPSAB1 and TPSB2 allele-specific genotyping will be very helpful.
Response: We thank this reviewer for the valuable feedback. According to the suggestion of the reviewer that methods for tryptase genotyping, in particular ddPCR, have been discussed in more detail in the revised version of our manuscript (line 418 to 427).
Mechanisms associated with mastocytosis and hereditary α-tryptasemia, the authors' thoughts, ideas, perspectives gained from previous research are valuable to provide more insight to the readers. Those studies that examined the influence of hereditary alpha tryptasemia on mast cell activation including those that tried to elucidate or hypothesize the molecular differences between the mutat KIT and WT KIT receptor and the impact on their localization, function, and susceptibility to phosphatases, etc are interesting determinants that need to be included. Specific details that could provide any insight on the mechanism by which gain in TPSABP1 copy number can lead to the mastocytosis and the clinical features of mastocytosis will be of added value. More details on how does mast cell tryptase serve as a potential target and additional descriptions about the potential mechanisms by which anti-tryptase antibody can block anaphylaxis/severe asthma in patients with mastocytosis could provide more interest to readers. What about patients with hereditary α-tryptasemia but failed to show any evidence of mastocytosis?
Response: We agree that the mechanisms of the association of mastocytosis and hereditary α-tryptasemia are highly interesting. The key question how on the mechanism by which gain in TPSABP1 copy number can lead to the mastocytosis and the clinical features of mastocytosis is currently largely unclear and subject of ongoing studies. We added a brief discussion on potential mechanisms and planned experiments in the revised version of our manuscript (line 640 to 647). However, as we are not aware of any data on the effect of TPSABP1 copy number gain on KIT D816V vs. WT signaling, we restrained from detailed speculation on this point. In addition, we included a discussion of the potential mechanisms of anti-tryptase antibodies in mastocytosis and hereditary α-tryptasemia (line 682 to 691).
The scope of developing genetic biomarkers by identifying the gain in TPSAB1 copy number for anaphylaxis risk prediction in patients with mastocytosis should be elaborately discussed.
Response: We thank this reviewer for addressing this point. Indeed, our data and additional publications indicate that TPSAB1 copy number gain is a risk factor for anaphylaxis in patients with mastocytosis. Further studies are required to demonstrate how TPSAB1 CNV based risk prediction can be incorporated into clinical practice and if TPSAB1 gain is a biomarker predictive for specific therapeutic interventions. A discussion of this important point has been included in the revised version of our manuscript according to the suggestion of the reviewer (line 621 to 630).
Minor points
page 7 line 306: by immunohistochemically stains.
Response: This point has been changed accordingly in the revised version of our manuscript (line 313).
Reviewer 2 Report
This is a comprehensive summary of research developments contributing to the evolving appreciation of contributions by genetic variations in mast cell tryptases to human syndromes and diseases, including allergy, mast cell disorders and myeloid malignancies. The summary is mostly accurate and balanced. I have the following suggestions for improvement:
Lines 107-110: Clarify what is meant by beta-heterotetramer. Did you mean beta-homotetramer? Or alpha/beta heterotetramer? Or a tetramer composed of different beta (I,II,or III) protomers? Each of these is thought to occur in humans (see in particular your ref. 188) with variations depending on pattern of inheritance (or lack thereof) of particular alpha and beta alleles at TPSAB1 and TPSB2.
Lines 113-115: The importance of sequential autocatalytic then dipeptidyl peptidase I-mediated processing of pro alpha and beta to mature tryptases has been strongly challenged by more recent work (see two papers published in 2011 by Le QT et al J Immunol 186:7136 and J Immunol 187:1912) showing that monomeric pro-tryptases are matured directly to active tetramers by cathepsins B and L in human mast cells. This also may be true in mice, which produce active mast cell tryptase in genetically modified animals lacking dipeptidyl peptidase I (see Wolters PJ et al J Biol Chem 276:18551, 2001).
Line 119: Alpha-tryptase is not “nearly identical” to beta-II tryptase. At the amino acid level, identity is only about 92%, with ~20 amino acid differences in the catalytic domain, including differences in pro-peptide affecting the potential for autoactivation, the mentioned key mutation in the substrate-binding region greatly reducing catalytic capability, and also a mutation in beta eliminating one of two N-linked glycosylation sites, potentially affecting stability and other physicochemical properties.
Line 231: “TKI inhibitor” is redundant. Can change to just “TKI” (but needs to be defined).
Lines 237-239: Unless I missed something, the cited studies by Sperr et al do not use an immunoassay capable of distinguishing pro-alpha from pro-beta. Although it was originally suspected that most protryptase shed “constantly” by mast cells and present in the circulation is pro-alpha, this is now known not to be the case because 1) mast cells studied in vitro spontaneously shed pro-beta along with pro-alpha and 2) because absence of alpha genes has little effect on basal level of serum pro-tryptase (see for example Akin C et al Clin Immunol 123:268, 2007).
Lines 274-276: The assumption that anaphylactic release of tryptase from mast cell granules is all beta may be inaccurate given 1) that the antibodies used in the assays do not distinguish between mature alpha and beta and 2) evidence that mast cells in humans inheriting alpha and beta genes form mature alpha and beta homo- and hetero-tetramers (your ref. 188).
Line 306: Change “immunohistochemically” to “immunohistochemical”
Lines 351-354: To my knowledge, the first complete sequence and structure of a human tryptase gene (rather than cDNA) was reported by Vanderslice et al (your ref. 172). And the first detailed BAC-based maps of the multi-gene human cluster were reported by Pallaoro et al (your ref. 171), who also first localized the cluster to chromosome 16p13.3 by FISH.
Line 357 and 397-401: The mention and discussion of so-called epsilon-tryptase could easily be omitted because 1) it is not a mast cell product, 2) it has different names as accepted by UniProt (Brain-specific protease-4) and MEROPS (prosemin), 3) it is only distantly related to mast cell alpha, beta and delta tryptases (being much more closely related to non-tryptases marapsin/PRSS27 and prostasin/PRSS8), and 4) the epsilon tryptase/prosemin/BSSP4/PRSS22 gene is not part of the alpha/beta/delta/gamma cluster, being megabases distant with many non-protease genes separating PRSS22 from the cluster.
Lines 368-373: there are inaccuracies here. There are multiple amino acid differences in the catalytic domain between betaIII and both betaI and betaII. The TPSAB1 alpha and beta (I) alleles have numerous dissimilarities in addition to those mentioned, especially in exons (and not limited to exon 3), leading to about 20 amino acid differences in the protein products of the respective alleles.
Line 510: To my knowledge, the proportion of alpha vs beta contributing to observed elevations of immunoreactive serum tryptase in “Hereditary alpha-tryptasemia” remains to be formally established, although it is reasonable to hypothesize that alpha is the major contributor due to a gene dosage effect.
Line 533-534: Ref. 188 does not seem to report studies on PAR2 in HUVECs.
Author Response
Reviewer 2:
This is a comprehensive summary of research developments contributing to the evolving appreciation of contributions by genetic variations in mast cell tryptases to human syndromes and diseases, including allergy, mast cell disorders and myeloid malignancies. The summary is mostly accurate and balanced. I have the following suggestions for improvement:
Lines 107-110: Clarify what is meant by beta-heterotetramer. Did you mean beta-homotetramer? Or alpha/beta heterotetramer? Or a tetramer composed of different beta (I,II,or III) protomers? Each of these is thought to occur in humans (see in particular your ref. 188) with variations depending on pattern of inheritance (or lack thereof) of particular alpha and beta alleles at TPSAB1 and TPSB2.
Response: We thank this reviewer for the detailed and valuable feedback that substantially improved our manuscript. The term β-heterotetramer was indeed erroneously used and has been changed to α/β-tryptase heterotetramers (line 107).
Lines 113-115: The importance of sequential autocatalytic then dipeptidyl peptidase I-mediated processing of pro alpha and beta to mature tryptases has been strongly challenged by more recent work (see two papers published in 2011 by Le QT et al J Immunol 186:7136 and J Immunol 187:1912) showing that monomeric pro-tryptases are matured directly to active tetramers by cathepsins B and L in human mast cells. This also may be true in mice, which produce active mast cell tryptase in genetically modified animals lacking dipeptidyl peptidase I (see Wolters PJ et al J Biol Chem 276:18551, 2001).
Response: We agree that this point has been challenged and included the data on tryptase maturation by cathepsins in the revised version of our manuscript (line 115 to 118).
Line 119: Alpha-tryptase is not “nearly identical” to beta-II tryptase. At the amino acid level, identity is only about 92%, with ~20 amino acid differences in the catalytic domain, including differences in pro-peptide affecting the potential for autoactivation, the mentioned key mutation in the substrate-binding region greatly reducing catalytic capability, and also a mutation in beta eliminating one of two N-linked glycosylation sites, potentially affecting stability and other physicochemical properties.
Response: We completely agree that relevant functional differences between alpha and beta-II are present and that the term “nearly identical” may therefore be misleading. We were primarily referring to the homology of the alleles on DNA level that made it difficult to differentiate the alleles in SNP-array-based GWAS studies. This explains why specific molecular techniques were required to study the association of hereditary α-tryptasemia and mastocytosis and the connection of was not. We rephrased this paragraph accordingly to make this point clear and omitted the term “nearly identical” in the revised version of our manuscript (line 122 to 129; see also response to Lines 374-382).
Line 231: “TKI inhibitor” is redundant. Can change to just “TKI” (but needs to be defined).
Response: We agree and corrected the term to tyrosine kinase inhibitor in our revised manuscript (line 238).
Lines 237-239: Unless I missed something, the cited studies by Sperr et al do not use an immunoassay capable of distinguishing pro-alpha from pro-beta. Although it was originally suspected that most protryptase shed “constantly” by mast cells and present in the circulation is pro-alpha, this is now known not to be the case because 1) mast cells studied in vitro spontaneously shed pro-beta along with pro-alpha and 2) because absence of alpha genes has little effect on basal level of serum pro-tryptase (see for example Akin C et al Clin Immunol 123:268, 2007).
Response: We agree the immunoassay used by Sperr et al. did not distinguishing pro-alpha from pro-beta tryptase. We thank the reviewer for raising this point and rephrased the paragraph accordingly (line 244 to 245).
Lines 274-276: The assumption that anaphylactic release of tryptase from mast cell granules is all beta may be inaccurate given 1) that the antibodies used in the assays do not distinguish between mature alpha and beta and 2) evidence that mast cells in humans inheriting alpha and beta genes form mature alpha and beta homo- and hetero-tetramers (your ref. 188).
Response: We agree that mature alpha and beta homo- and hetero-tetramers may contribute to the anaphylactic release of tryptase in individuals inheriting alpha genes. We thank the reviewer for this valuable comment and rephrased the paragraph accordingly (“(resulting from β-tryptase release in MCs)” in line 273 and “β-tryptase” in line 280).
Line 306: Change “immunohistochemically” to “immunohistochemical”
Response: This point has been changed accordingly in the revised version of our manuscript (line 313).
Lines 351-354: To my knowledge, the first complete sequence and structure of a human tryptase gene (rather than cDNA) was reported by Vanderslice et al (your ref. 172). And the first detailed BAC-based maps of the multi-gene human cluster were reported by Pallaoro et al (your ref. 171), who also first localized the cluster to chromosome 16p13.3 by FISH.
Response: We thank this reviewer for clarifying this point and rephrased the description of the discovery of human tryptase genes accordingly (line 358 to 363).
Line 357 and 397-401: The mention and discussion of so-called epsilon-tryptase could easily be omitted because 1) it is not a mast cell product, 2) it has different names as accepted by UniProt (Brain-specific protease-4) and MEROPS (prosemin), 3) it is only distantly related to mast cell alpha, beta and delta tryptases (being much more closely related to non-tryptases marapsin/PRSS27 and prostasin/PRSS8), and 4) the epsilon tryptase/prosemin/BSSP4/PRSS22 gene is not part of the alpha/beta/delta/gamma cluster, being megabases distant with many non-protease genes separating PRSS22 from the cluster.
Response: We thank epsilon-tryptase is not relevant for the scope of the review and omitted the discussion of this gene according to the suggestion of the reviewer (text passages deleted in line 364, 369 and 406).
Lines 368-373: there are inaccuracies here. There are multiple amino acid differences in the catalytic domain between betaIII and both betaI and betaII. The TPSAB1 alpha and beta (I) alleles have numerous dissimilarities in addition to those mentioned, especially in exons (and not limited to exon 3), leading to about 20 amino acid differences in the protein products of the respective alleles.
Response: We completely agree that relevant functional differences between alpha and beta tryptases are present and that the wording may therefore have been misleading (see also Response to Line 122 to 129). We rephrased this paragraph accordingly in the revised version of our manuscript to make this point clear and to avoid the appearance of a lack of functional differences between the tryptases (line 374 to 382).
Line 510: To my knowledge, the proportion of alpha vs beta contributing to observed elevations of immunoreactive serum tryptase in “Hereditary alpha-tryptasemia” remains to be formally established, although it is reasonable to hypothesize that alpha is the major contributor due to a gene dosage effect.
Response: We agree that this concept needs to be formally established and changed the phrasing accordingly (line 523 to 525).
Line 533-534: Ref. 188 does not seem to report studies on PAR2 in HUVECs.
Response: We thank the reviewer who pointed out the discrepancy and corrected the citation accordingly (line 565 to 571; the correct study is from Lyons JJ et al. J Allergy Clin Immunol 2020).
Reviewer 3 Report
This manuscript is informative and is generally written well with rigor. The authors provided detailed information about mast cell tryptase and covered the most recent research articles as well. The text is presented clearly, and logical flow justifies the authors' points.
No point needs to be addressed.
Author Response
Reviewer 3:
This manuscript is informative and is generally written well with rigor. The authors provided detailed information about mast cell tryptase and covered the most recent research articles as well. The text is presented clearly, and logical flow justifies the authors' points.
No point needs to be addressed.
Response: We thank this reviewer for the encouraging feedback.